# Reducing bias in coronary heart disease prediction using Smote-ENN and PCA

**Xinyi Wei**[1]*, **Boyu Shi**[2]

**1** Universiti Malaya, Institute for Advanced Studies, Universiti Malaya, Kuala Lumpur, Wilayah Persekutuan Kuala Lumpur, Malaysia, **2** Universiti Malaya, Academy of Islamic Studies, Universiti Malaya, Kuala Lumpur, Wilayah Persekutuan Kuala Lumpur, Malaysia

* wxye1848306171@outlook.com

## Abstract

Coronary heart disease (CHD) is a major cardiovascular disorder that poses significant threats to global health and is increasingly affecting younger populations. Its treatment and prevention face challenges such as high costs, prolonged recovery periods, and limited efficacy of traditional methods. Additionally, the complexity of diagnostic indicators and the global shortage of medical professionals further complicate accurate diagnosis. This study employs machine learning techniques to analyze CHD-related pathogenic factors and proposes an efficient diagnostic and predictive framework. To address the data imbalance issue, SMOTE-ENN is utilized, and five machine learning algorithms—Decision Trees, KNN, SVM, XGBoost, and Random Forest—are applied for classification tasks. Principal Component Analysis (PCA) and Grid Search are used to optimize the models, with evaluation metrics including accuracy, precision, recall, F1-score, and AUC. According to the random forest model's optimization experiment, the initial unbalanced data's accuracy was 85.26%, and the F1-score was 12.58%. The accuracy increased to 92.16% and the F1-score reached 93.85% after using SMOTE-ENN for data balancing, which is an increase of 6.90% and 81.27%, respectively; the model accuracy increased to 97.91% and the F1-score increased to 97.88% after adding PCA feature dimensionality reduction processing, which is an increase of 5.75% and 4.03%, respectively, compared with the SMOTE-ENN stage. This indicates that combining data balancing and feature dimensionality reduction techniques significantly improves model accuracy and makes the random forest model the best model. This study provides an efficient diagnostic tool for CHD, alleviates the challenges posed by limited medical resources, and offers a scientific foundation for precise prevention and intervention strategies.

**Data availability statement:** All relevant data are within the paper.

**Funding:** The author(s) received no specific funding for this work.

**Competing interests:** The authors have declared that no competing interests exist.

## Introduction

Cardiovascular disease (CVD) remains the leading cause of death globally, particularly in low- and middle-income countries [1]. Among CVD types, coronary heart disease (CHD) is the most common, characterized by myocardial ischemia and necrosis caused by coronary artery atherosclerosis [2]. Over the past two decades, the number of CVD patients has surged to 520 million worldwide [3]. Unhealthy lifestyles such as smoking, alcohol consumption, and high-sugar diets are major risk factors, increasingly prevalent among younger populations [4]. The COVID-19 pandemic further exposed the vulnerability of individuals with heart conditions [5].

Additionally, environmental pollution, genetic predisposition, and comorbidities exacerbate the complexity of CHD. Prolonged exposure to air pollution can induce systemic inflammation and autonomic nervous system imbalance, increasing the risk of stroke and CHD [6]. Genetic studies have identified specific gene polymorphisms closely related to CHD [7]. Psychological traits, such as conscientiousness and neuroticism, can also affect disease outcomes [8]. Early diagnosis and prevention of CHD thus require multidisciplinary support and improved prediction models.

However, traditional CHD diagnosis methods face challenges related to prediction accuracy and applicability across diverse populations [9]. Machine learning (ML) models, leveraging mathematical and statistical frameworks, have shown great promise in disease identification and diagnostic support. Models such as Support Vector Machines (SVM), Random Forest, and Logistic Regression have demonstrated high accuracy and robustness in medical prediction. Yet, data imbalance and prediction bias remain critical challenges that need to be addressed.

To mitigate these issues, researchers have explored techniques such as SMOTE-ENN (Synthetic Minority Oversampling Technique – Edited Nearest Neighbors) to balance datasets [10]. These methods are crucial for reducing bias and improving model prediction accuracy. Furthermore, analyzing correlations between various medical features is essential for identifying key CHD risk factors and understanding disease mechanisms [11]. Employing grid search and principal component analysis (PCA) for dimensionality reduction allows models to be evaluated based on multiple performance metrics, including accuracy, precision, recall, F1 score, and the area under the ROC curve (AUC).

Imbalanced datasets pose significant challenges in fields such as image segmentation, deep learning, fraud detection, data mining, and intrusion detection [12–15]. Accordingly, several strategies have been proposed to reduce bias and enhance model robustness [16,17].

In medical diagnosis, particularly CHD prediction, SMOTE-ENN has emerged as a key solution for addressing data imbalance. It generates synthetic samples and removes noisy data, significantly improving prediction performance. In clinical decision support systems, it enhances model accuracy [18], boosts bankruptcy prediction models in financial analytics [19], and optimizes student performance prediction in educational data analysis [20].

Thus, this study aims to integrate these successful practices to optimize CHD prediction models by addressing the negative impacts of data imbalance and prediction bias on model performance. This approach supports improved medical diagnosis and broader machine learning applications, leading to better predictive outcomes [21–24].

In summary, this paper focuses on optimizing CHD prediction models by applying SMOTE-ENN to balance imbalanced datasets and reduce model prediction bias. Additionally, combining grid search and PCA optimizes model parameters for machine learning algorithms such as Decision Trees, KNN, SVM, XGBoost, and Random Forest. This comprehensive approach aims to identify major CHD risk factors, build a more accurate and stable medical diagnostic support system, and facilitate early prevention and personalized treatment.

## Related work

In recent years, research on coronary heart disease (CHD) has primarily focused on two key areas: data imbalance studies and the development of auxiliary diagnostic models. These two directions aim to address critical challenges in CHD prediction model development, thereby improving model accuracy and reliability.

Due to the naturally uneven distribution of positive and negative samples in medical datasets, data imbalance is particularly common in CHD prediction research. This imbalance significantly affects model performance, leading to prediction bias and higher misclassification rates. To address this issue, researchers have applied various resampling techniques such as SMOTE, SMOTE-ENN, and undersampling to balance data distribution and improve model performance. The research progress is summarized in Table 1.

At the same time, CHD diagnostic research has increasingly emphasized the use of various machine learning algorithms to develop auxiliary diagnostic models, enhancing early detection and decision-making support for CHD. These models improve diagnostic accuracy and stability through feature selection, optimization, and predictive modeling. The research progress is summarized in Table 2.

## Data processing and visualization

The dataset used in this dissertation's investigation of coronary heart disease is sourced from the "framingham.csv" database on Kaggle. It originates from an ongoing cardiovascular study involving residents of Framingham, Massachusetts. The primary goal of data collection is to predict whether individuals are likely to develop coronary heart disease within the next 10 years. The dataset comprises 4,238 samples and 16 variables, including one target variable and 15 feature variables. The target variable is used to forecast whether participants will experience coronary heart disease within a ten-year period. As shown in Table 3, In this experiment, non-numeric data needs to be converted to numeric data because numeric data. The dataset used in this project contains 15 feature variables and 1 target variable, of which 6 are binary variables, 1 is a multi-category variable, and 8 are numeric variables.

**Table 1. Summary Table.**

| Study | Data Source | Data Size | Methods Used | Accuracy |
|---|---|---|---|---|
| Wu (2022) [25] | Suzhou, China | Not Specified | AdaBoost, Association Rule Mining | Not Reported |
| Su et al. (2022) [26] | Kaggle | Not Specified | SMOTE, Ensemble Algorithms (Random Forest, XGBoost, LightGBM) | Not Reported |
| Minou et al. (2020) [27] | UCI Repository | Not Specified | SMOTE, Various Classifiers | Not Reported |
| Zhang (2023) [28] | Kaggle | Not Specified | SMOTE-ENN, Stacking Model (Random Forest, AdaBoost, LightGBM, KNN) | Not Reported |
| Trigka & Dritsas (2023) [29] | Not Specified | Not Specified | SMOTE, Stacking Ensemble Model | 90.9% |

**Table 2. Summary Table.**

| Study | Data Source | Methods Used | Key Outcomes |
|---|---|---|---|
| Zhu et al. (2013) [30] continuation | Physical examination markers | PSO with SVM variants (RBF, Polynomial, Linear) | PSO-RBF-SVM showed best performance; exceeded 92.31% in precision, specificity, sensitivity. |
| Zeinab et al. (2017) [31] | CHD diagnostic dataset | Hybrid method (neural networks and genetic algorithms) | Improved neural network performance and CHD detection accuracy. |
| Krishnani et al. (2019) [32] | Framingham Heart Study dataset | K-nearest neighbour, random forest, decision tree | Random forest showed the most accurate results. |
| Yin (2019) [33] | Data from a tertiary grade A hospital | SVM with seven feature selection techniques | Integrated feature selection method enhanced SVM's reliability in CHD identification. |
| Charles Bemando et al. (2021) [34] | UCI Cleveland Database | Gaussian naive Bayes, Bernoulli naive Bayes, random forest | Gaussian and Bernoulli naive Bayes showed 85% accuracy, higher precision, F-measure, and recall than random forest. |
| Li et al. (2020) [35] | Kaggle CHD data | Logistic regression, SVM, LDA, decision tree, random forest | Identified key risk factors; models demonstrated good accuracy and stability. |
| Mangathayaru et al. (2020) [36] | Not specified | Linear models, naive Bayes, CART, SVM, KNN, ensemble models | Provided insights for CHD decision support systems. |
| Krittanawong et al. (2020) [37] | Not specified | Various ML algorithms | Demonstrated variability in results using different methods; meta-analysis improved predictions. |
| Hong et al. (2023) [38] | Kaggle CHD dataset | Logistic regression, XGBoost | XGBoost outperformed logistic regression in accuracy, recall, AUC. |
| Lakshmi Padmaja et al. (2022) [39] | Kaggle CHD data | Logistic regression, random forest, KNN, SVM, decision tree | Used for early prediction of CHD. |
| continuation Alhammadi et al. (2023) [40] | Framingham and Cleveland Heart Disease datasets | Naive Bayes, decision trees, SVM, KNN | Decision tree trained on the Cleveland dataset showed highest accuracy. |

## A. Data reduction

Since many existing algorithms do not support missing value inputs, it is essential to address missing data before training. The presence of missing values indicates a significant loss of relevant information, increasing the overall uncertainty of the dataset. By applying missing value imputation techniques, the true data can be partially reconstructed, providing valuable insights and enhancing the model's performance.

Data analysis reveals that the following feature variables contain missing values: Education, Cigarettes per Day (cigsPerDay), Use of Blood Pressure Medication (BPMeds), Total Cholesterol (totChol), Body Mass Index (BMI), Heart Rate (heartRate), and Glucose Level (glucose). Table 4 presents the number and percentage of missing values for each feature. The missing value of the glucose level characteristic is close to 10%, which will have a significant impact on the diagnostic performance of the model.

Missing values of variables complicate the data analysis process and may skew the final results [24]. This thesis uses the common mode filling method to process missing values based on label data such as education and male. For cigsPerDay, BPMeds, totChol, BMI, heartRate, and glucose, these feature variables are numerical values that are filled in using the mean-filling approach. The goal of the processing described in this research is to keep as much data as feasible while preserving the statistical properties of the data. Specifically, for classification features (such as education and male), the mode is used to fill in the missing values. For numerical features (such as cigsPerDay, BPMeds, etc.), use the mean to fill in the missing values.

**Table 3. Description of the attributes of the dataset.**

| Type | Category | Description | Probable Value |
|---|---|---|---|
| Nominal | Male | Male or female patient | male (0),female(1) |
| | currentSmoker | Whether the patient is a current smoker or not | No = 0, Yes = 1 |
| | BPMeds | Whether the patient was taking blood pressure medication. | |
| | prevalentStroke | Whether the patient has had a previous stroke | |
| | prevalentHyp | Whether the patient has had a previous hypertensive | |
| | diabetes | Whether the patient has had diabetes | |
| | TenYearCHD | 10-year risk of coronary heart disease in the patient | |
| Continous | Age | The patient's age | 29-64 |
| | cigsPerDay | The average number of cigarettes smoked by the individual in a single day | 0 day-70 days |
| | totChol | Total cholesterol level in the patient | 107mg/dL-696 mg/dL |
| | sysBP | Systolic blood pressure of the patient | 83.5mmHg-295 mmHg |
| | diaBP | Diastolic blood pressure of the patient | 48mmHg-142.5 mmHg |
| | BMI | Patient Body Mass Index Weight(kg)/Height(meter-squared) | 15.54 kg/m²-56.8 kg/m² |
| | heartRate | Patient heart rate | 44bpm-143bpm |
| | glucose | Patient glucose level | 40mg/dL-394 mg/dL |
| Categorical | education | Level of patient education | high school(1), high school or GED (2), college or vocational school(3), college(4) |

**Table 4. Missing values and percentages for each attribute.**

| Attribute | Number of missing values | Percentage of missing values(%) |
|---|---|---|
| education | 105 | 2.48 |
| cigsPerDay | 29 | 0.68 |
| BPMeds | 53 | 1.25 |
| totChol | 50 | 1.18 |
| BMI | 19 | 0.45 |
| heartRate | 1 | 0.02 |
| glucose | 388 | 9.15 |

## B. Data Discretization

**a.Data standardization.** Data normalization is a sort of data preparation in which the values of different characteristics are scaled to a similar range in order to better adapt to the various machine learning algorithm models that follow. The primary goal of data standardization is to remove dimensional disparities between features in order to accelerate model convergence and improve model performance.

The Z-score of a feature can be computed using the following formula based on the characteristics of this experimental data, the standard normal distribution with a standard deviation of 1 (also known as Z-score normalisation):

$$Z = \frac{X - \mu}{\sigma}$$

(1)

Among them, *Z* is the normalized value. *X* is the initial value. *μ* is the feature's mean (mean). σ is the fea ture's standard deviation.

**b.Data normalization.** Data normalisation is also a component of data preprocessing. Its function is to scale data to a given range. This experiment employs values ranging from 0 to 1.

Data normalisation is typically accomplished by applying the following formula, which translates the data to the range [0, 1]. The formula for data normalisation is as follows:

$$X_{normalized} = \frac{X - X_{min}}{X_{max} - X_{min}}$$

(2)

Among them, $X_{normalized}$ is the normalized value. X is the original value. $X_{min}$ is the feature's minimum value. $X_{max}$ is the feature's maximum value.

For coronary heart disease data, it is beneficial to standardize and normalize the data because the data set includes features of different units and ranges, such as cigsPerDay, totChol, BMI, heartRate, etc. Through data standardization and normalization, all these features can be scaled to similar scales, which helps the model handle these features better and improves the performance and convergence speed of the model.

**C.Feature selection.** The purpose of feature selection is to identify feature variables that contribute significantly to the model by examining the relationship between features in order to improve the model's performance.

The coronary heart disease data in this experiment contains a large number of characteristics, some of which may not be useful in predicting coronary heart disease and may even contribute to noise. The goal of feature selection is to decrease dimensionality and the risk of overfitting by removing the characteristics that are most important for predicting coronary heart disease.

In this case study, the filtering strategy for feature selection is used. A common filtering technique is to use statistical markers such as variance, mutual information, and the chi-square test (CHI2) to evaluate the relationship between characteristics and labels [41–42]. In this inquiry, the chi-square test will be utilised to pick characteristics. The chi-square test is used to detect if two variables in a category are related or independent. In this experiment, we have two variables: characteristics (X) and aim variables (y).

The Chi-square test was used to evaluate the connection between the target variable (TenYearCHD) and each feature. The chi-square test is carried out in the following manner: Create a contingency table using the observed values of the characteristics and target variables. The predicted frequency for each cell is then computed, which is the frequency of observations in that cell, assuming that the feature and target variable are independent. Finally, the chi-square statistic is calculated using the formula:

$$X^2 = \sum \frac{(O - E)^2}{E}$$

(3)

where $X^2$ is the chi-square statistic, O is the observed frequency, and E is the expected frequency.

The chi-square statistic for the whole contingency table is obtained by adding all cells together. A higher chi-square statistic score suggests that the characteristic predicts the target variable better. Fig 1 (a) illustrates each feature's relevance score from the table below. On the coronary heart disease data set, the correlation between each feature and the risk of coronary heart disease was evaluated, and the chi-square test was used to rank the value of the characteristics to achieve the purpose of quickly screening features. Perform a chi-square test on the data set's 15 category feature variables using Python. Table 5 and Fig 1(b) displays the results of the chi-square test on the characteristic variable and the following conclusions.

## D.Dimensionality reduction

Dimensionality reduction simplifies a dataset by reducing its attributes while preserving critical information, lowering processing costs, and enhancing model efficiency. Common techniques include Principal Component Analysis (PCA), Linear

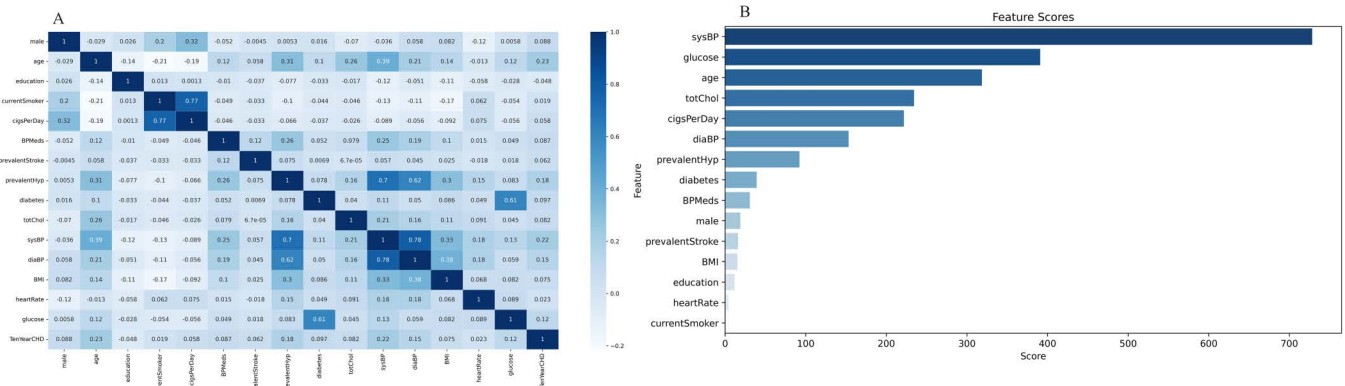

**Fig 1. Heat maps and results of feature importance. a.** Heat map of correlation of each feature, **b.** Importance score of each feature.

**Table 5. Characteristic variable scores for the chi-square test.**

| Feature | Conclusion | Score |
|---|---|---|
| sysBP | The crucial conclusion is that systolic blood pressure has a significant role in predicting the probability of coronary heart disease. Increased systolic blood pressure is usually associated with cardiovascular problems. | 728.292522 |
| glucose | Diabetes, a known risk factor for coronary heart disease, is indicated by high glucose levels. | 390.825416 |
| age | As the risk of cardiovascular disease grows with age, this is an important element in risk prediction. | 318.597444 |
| totChol | It is a key predictor of coronary heart disease due to its close link with cardiovascular health. | 234.305978 |
| cigsPerDay | Consider the potential negative health impacts of smoking, which has been linked to an increased risk of heart disease. | 221.680719 |
| diaBP | Although somewhat less relevant than systolic blood pressure, diastolic blood pressure is nevertheless an essential blood pressure measure in prediction models. | 153.090954 |
| prevalentHyp | Hypertension is a prominent risk factor for cardiovascular disease, and the feature scores show that it has a role in the model's prediction. | 92.167857 |
| diabetes | Diabetes is related to a variety of cardiovascular problems and is a key factor in predicting coronary heart disease risk, as evidenced by trait scores. | 39.103710 |
| BPMeds | 1.It might be a hypertensive response, or it could be an indication of cardiovascular disease. 2. It is a risk factor for coronary artery disease, and men are more vulnerable than women. | 30.578341 |
| male | A history of stroke implies that a person is more likely to develop cardiovascular problems. | 18.915212 |
| prevalentStroke | The model's importance score is low, yet it is routinely used to analyse the influence of weight on health, particularly the risk of cardiovascular disease. | 16.095638 |
| BMI | Although educational level is related to health behaviours and outcomes, it does not appear to be a substantial predictor in our model. | 15.177679 |
| education | Although heart rate is an important measure of cardiovascular health, it plays little role in this model. | 6.271609 |
| heartRate | The model gave less weight to whether or not a person smoked, suggesting that daily cigarette intake is more powerful than simple smoking habits in predicting the risk of coronary heart disease. | 4.2334763 |
| currentSmoker | Diabetes, a known risk factor for coronary heart disease, isindicated by high glucose levels. | 0.811603 |

Discriminant Analysis (LDA), t-distributed Stochastic Neighbor Embedding (t-SNE), and feature selection methods like Recursive Feature Elimination.

This research applies PCA to reduce the dimensionality of the coronary heart disease dataset. PCA transforms original data into linearly uncorrelated principal components, capturing most data variance with fewer variables. Its benefits include reducing feature numbers, simplifying data structure, improving analysis efficiency, lowering computational costs,

speeding up model training and prediction, minimizing data noise, and enhancing prediction accuracy. As shown in Fig 2, we conducted PCA visualization. The variance interpretation rates of each principal component can be understood through the visualization results. In Fig 2, the left figure shows the visualization results of PC1-PC2, and the right figure shows the visualization results of PC3-PC4. It can be found that the sum of the variance explanations from PC1 to PC4 is 88%(> 85%). However, the specific dimension to which it should be reduced is related to the model. Therefore, the specific dimension situation will be explained in the subsequent experiments.

### Predictive Coronary Heart Disease Classification Model

Fig 3 illustrates the entire process from data collection to the evaluation of the coronary heart disease prediction model.

**Step 1:** The framework begins by obtaining coronary heart disease-related data from Kaggle and doing basic analysis on it.

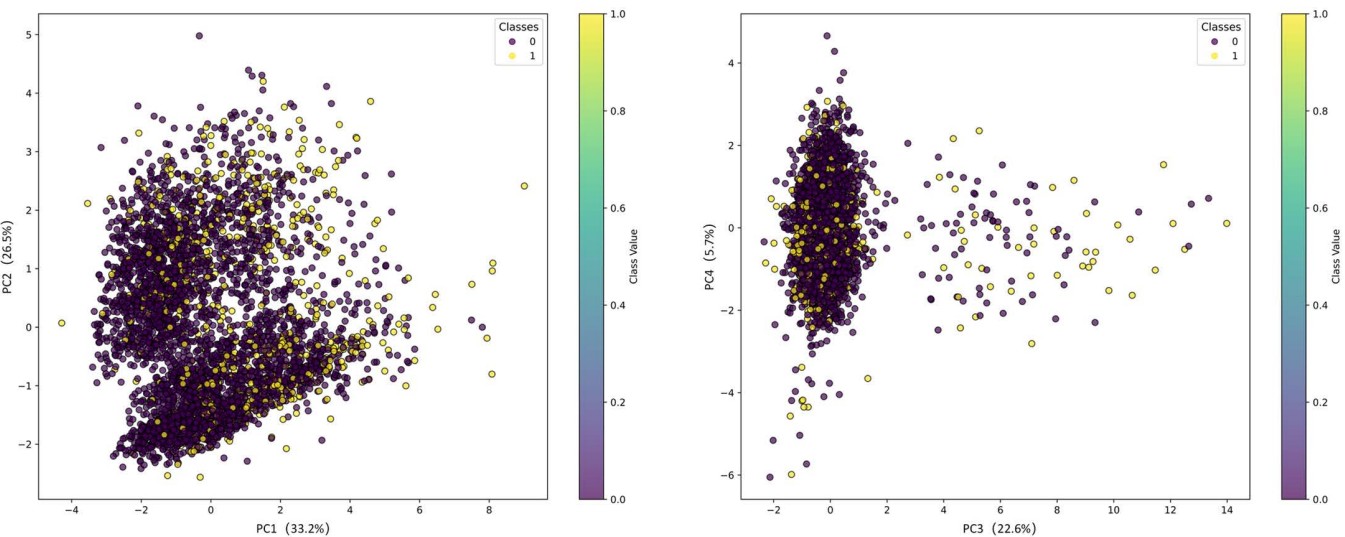

**Fig 2. Visualization of PCA results.**

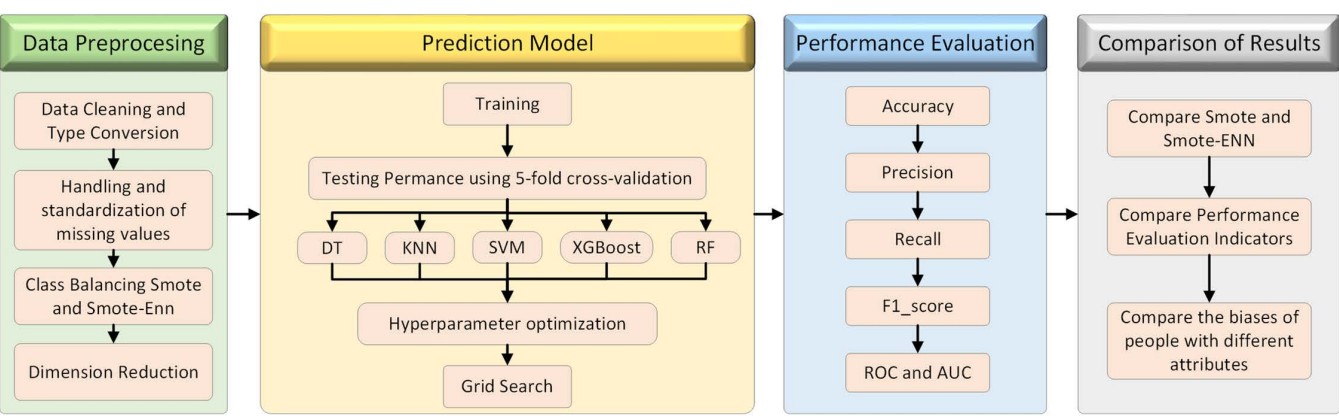

**Fig 3. Coronary heart disease framework diagram.**

**Step 2:** data preparation, which involves data type conversion. To build a basis for analysis, convert data into a model-ready format.

Data cleansing is the first cleaning of data to eliminate inconsistent or erroneous records.

Missing values must be handled carefully to guarantee that the analysis findings are accurate.

Data standardisation and normalisation: Standardisation eliminates the influence of differing dimensions data, allowing the model to assess every attribute more accurately.

Class balancing Smote and Smote-Enn: They are used to address data category imbalance and improve the model's prediction abilities for minority categories.

Feature Selection: To simplify the model and decrease the length of calculation, choose the features with the biggest influence on prediction outcomes from a large number of features.

Dimensionality reduction – PCA: Principal component analysis reduces the dimensionality of data, improving the model's computational efficiency and prediction speed.

**Step 3:** Prediction model

Training: Train different machine learning models on the processed data to discover the data's fundamental rules.

Use 5-fold cross-validation to test performance. This approach may evaluate the model's performance on unknown data while ensuring that the evaluation is fair and accurate.

Models include decision trees, KNN, SVM, XGBoost, and random forests, which span a wide range of methods from simple to complicated.

Optimisation: Use grid search to discover the parameter combination that best fits the data characteristics.

**Step 4:** Involves evaluating model performance using measures including accuracy, precision, recall, F1 score, ROC curves and AUC values are used to evaluate the model's prediction abilities, particularly when selecting categorization thresholds.

**Step 5:** Compare the findings.

Compare Smote to Smote-Enn, and investigate the implications of various data balancing techniques. To avoid bias, compare alternative performance evaluation metrics and assess the model's merits and shortcomings from many angles.

In short, this framework diagram shows that data collection and preparation precede model training and optimisation, followed by performance evaluation and outcome comparison. Every stage is critical to the accuracy and reliability of the study findings. This framework diagram not only outlines a specific study path for coronary heart disease data analysis, but it also gives a methodological framework that may be used to other comparable medical data analysis studies.

### A.SMOTE (oversampling)

Calculate the k nearest neighbours in the minority class sample set for each sample in the minority class.

Based on these nearest neighbours, choose a neighbour at random and then generate fresh sample points on the connection between the two points at random.

Repeat the preceding processes until the necessary number is obtained, effectively raising the number of minority class samples artificially. The formula is as follows:

$$X_{new} = X_i + \lambda \times (X_z - X_i) \tag{4}$$

Among them, X new is the new sample generated, $X_i$ is the original sample in the minority class, $\lambda$ is a random number between 0 and 1, and $X_z$ is a sample randomly selected from the k nearest neighbours of $X_i$.

### B.ENN (undersampling)

Use the closest neighbour methods to classify all samples. Examine each majority class sample's k nearest neighbours.

If the majority of the neighbours of a majority class sample are from the minority class, discard the majority class sample. This phase aids in the removal of majority class samples that could be noisy or easily misclassified around the decision boundary.

### C.SMOTE-ENN

SMOTEENN is a mixed sampling method that combines the SMOTE and ENN algorithms. To balance the data, new samples are generated using the SMOTE and SMOTE-ENN methods. The optimal method for balancing data during model training is determined by comparing changes in model indicators under the SMOTE and SMOTE-ENN algorithms.

## Experimental Results

In this paper, we explore decision trees, KNNs, SVMs, XGBoosts and random forests as five machine learning techniques used for predictive classification. The models were evaluated on accuracy, precision, recall, F1-score and AUC. The models were tuned using Principal Component Analysis (PCA) and grid search methods to reduce bias and improve their predictive accuracy for coronary heart disease. In addition, the predictive performance of the five machine learning algorithm models was compared and evaluated.

### A.Tools

In this study, Python 3.10.6 was selected as the primary data analysis tool. Python has become the preferred tool in the fields of data science and machine learning due to its extensive libraries and broad applications. Its powerful features include data retrieval, data monitoring and visualization, data statistics, handling missing values, data discretization, and feature selection. Moreover, Python supports flexible analysis and classification tasks, efficiently managing large datasets while ensuring the accuracy and reliability of the data analysis process. Therefore, in constructing and optimizing the coronary heart disease prediction model, Python provides comprehensive support for data preprocessing and model training, making it an indispensable analytical platform for this research.

### B.Comparison of Different Models

Without applying any balancing method, the Decision Tree algorithm achieved an AUC of 0.54, which is significantly lower compared to the AUCs of 0.81 and 0.83 obtained after applying the SMOTE and SMOTE-ENN balancing methods, respectively. This aligns with Brownlee's (2020) findings, which emphasized that Decision Tree algorithms generally perform better on balanced datasets. The increase in AUC from 0.54 to 0.83 also supports the results of Chawla et al. (2002), who reported that combining over-sampling and under-sampling techniques can lead to more robust models. The results are shown in Table 6. Among them, the min_samples_split in the decision tree is 2 and the min_samples_leaf is 1.

The confusion matrix and ROC curve displayed by the initial decision tree model on whether to use SMOTE-ENN are shown in Fig 4 and Fig 5.

With SMOTE-ENN, after employing grid search to optimise the model and using Principal Component Analysis (PCA) to process the data, PCA outperforms the grid search optimized model. In this case, n_components = 1. Table 7 presents the modeling results obtained using PCA and web search.

Table 6. Comparison of methods for balancing data in decision trees.

| Balance Method | Accuracy | precision | Recall | F1_score | AUC |
|---|---|---|---|---|---|
| Without | 74.76% | 20.78% | 25.81% | 23.02% | 0.54 |
| SMOTE | 81.36% | 78.63% | 83.67% | 81.07% | 0.81 |
| SMOTE-ENN | 85.65% | 86.75% | 90.48% | 88.57% | 0.83 |

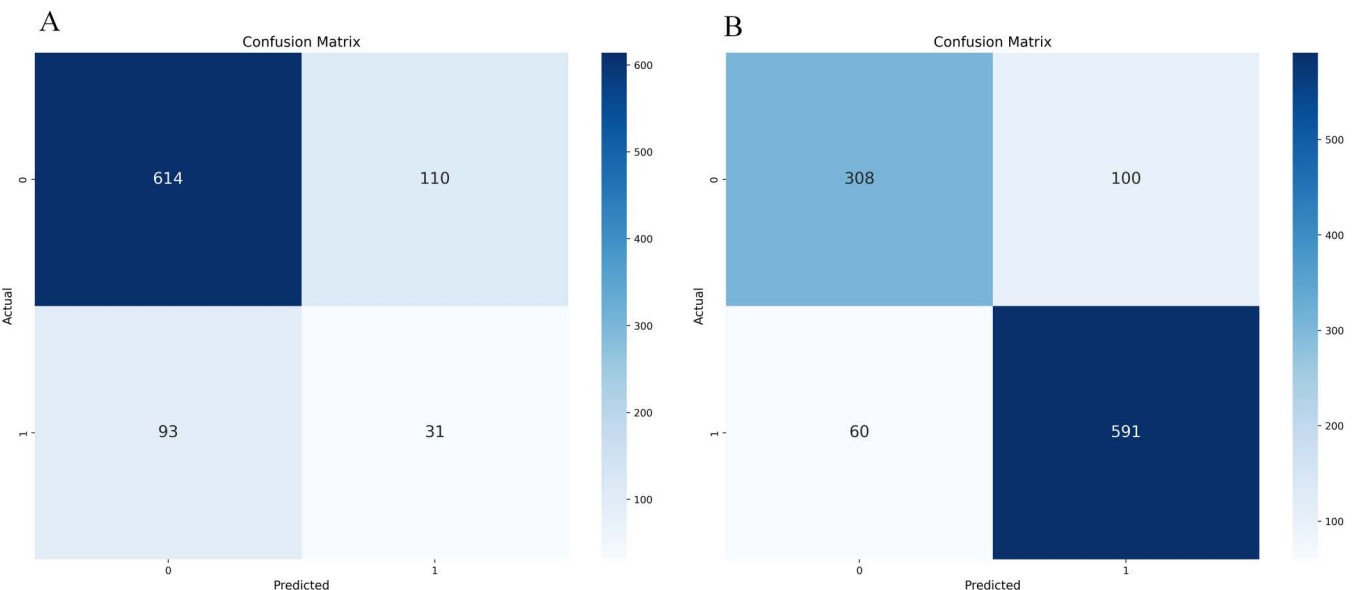

**Fig 4. Initialization confusion matrix of decision tree (with and without SMOTE-ENN). (a)** Without.**(b)** With SMOTE-ENN.

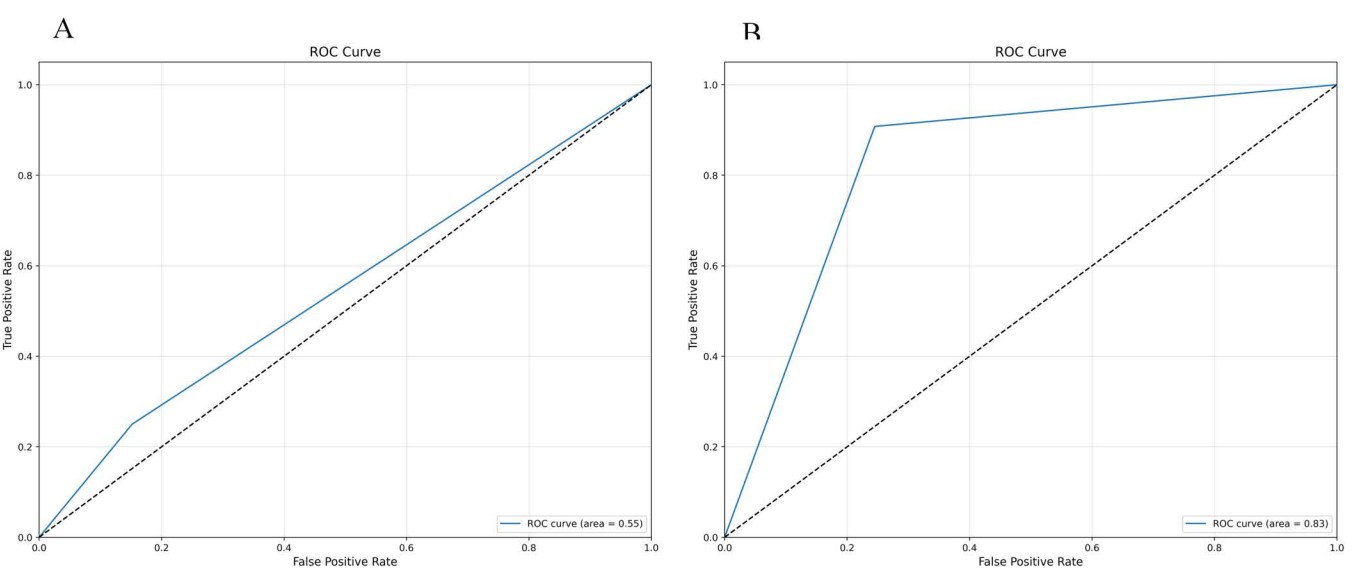

**Fig 5. Initialization ROC Plot of Decision Tree (with and without SMOTE-ENN). (a)** Without.**(b)** With SMOTE-ENN.

The model's prediction accuracy after PCA is 97.66%, meaning that it accurately predicts the overall percentage of positive and negative classes to be 97.66%. The high accuracy suggests that the model can provide accurate predictions in the majority of situations. With a precision of 96.97%, the model is generally correct in predicting that a patient has coronary heart disease. The model correctly detects virtually all patients with real coronary heart disease, with a recall rate of 98.25%. In order to account for both accuracy and recall, the F1 score is derived as the harmonic mean of these

**Table 7. Evaluation index after Decision Tree optimization parameters.**

| Method | Accuracy | Precision | Recall | F1-score | AUC |
|---|---|---|---|---|---|
| Grid search | 85.08% | 85.88% | 90.63% | 88.19% | 0.84 |
| PCA | 97.66% | 96.97% | 98.25% | 97.60% | 0.98 |

two criteria. The F1 score in this instance is likewise quite high, suggesting that the model has successfully struck a fair compromise between precision and recall, reducing the bias in predictions.

### C.Classification by k-nearest neighbors

KNN without balancing had a poor recall of 13.71%, which significantly improved to 99.23% with SMOTE-ENN. This is consistent with the work of Han et al. [43], where it was observed that KNN is sensitive to imbalanced data, and employing SMOTE-ENN can enhance the model's sensitivity to the minority class. The improvement in F1_score from 19.77% to 93.96% also resonates with the findings of Batista et al. [44] highlighting the effectiveness of SMOTE-ENN in improving classification performance in the presence of data imbalance. As shown in Table 8. Among them, the n_neighbors in KNN is 5, the leaf_size is 30, and p is 2.

The confusion matrix and ROC curve displayed by the initial KNN model whether to use SMOTE-ENN are shown in Figs 6–7.

With SMOTE-ENN, after using grid search to optimize the model and applying principal component analysis (PCA) to process the data, the grid search-optimized model outperforms PCA. As shown in Table 9. After grid search optimisation, the anticipated accuracy is 95%, precision is 92.84%, recall is 99.54%, and F1 is 96.078%. The accuracy, precision, recall, and F1-score have all increased by 2.84%, 3.61%, 0.31%, and 2.11%, respectively, when compared to the initial model. It shows that 95% of persons are accurately predicted in the optimised findings, while 99.54% of actual patients are appropriately forecasted as having the condition. Almost all cases can be correctly diagnosed using the optimised KNN model, displaying high predictive ability.

Overall, the model's performance improved after tuning due to better feature selection, optimal parameter settings, and appropriate complexity adjustments. The confusion matrix showed enhanced prediction accuracy for specific categories, while the AUC value from the ROC curve indicated better overall classification capability. Reducing false negatives (FN) is crucial in medical applications to prevent severe consequences from missed diagnoses. Additionally, the tuned KNN model successfully reduced false positives (FP), lowering the risk of misdiagnosis.

### D.Classification by support vector machines

For SVM, there is a notable improvement in recall from 2.42% with no balance to 92.63% with SMOTE-ENN. This demonstrates the susceptibility of SVM to class imbalance, as noted by Akbani et al. [45], and the potential of SMOTE-ENN to mitigate this issue. The precision also shows an improvement, which is in line with the observations by Tang et al. [46],

**Table 8. Comparison of methods for balancing data in KNN.**

| Balance Method | Accuracy | precision | Recall | F1_score | AUC |
|---|---|---|---|---|---|
| Without | 83.73% | 35.42% | 13.71% | 19.77% | 0.64 |
| SMOTE | 80.04% | 71.02% | 98.25% | 82.45% | 0.91 |
| SMOTE-ENN | 92.16% | 89.23% | 99.23% | 93.96% | 0.98 |

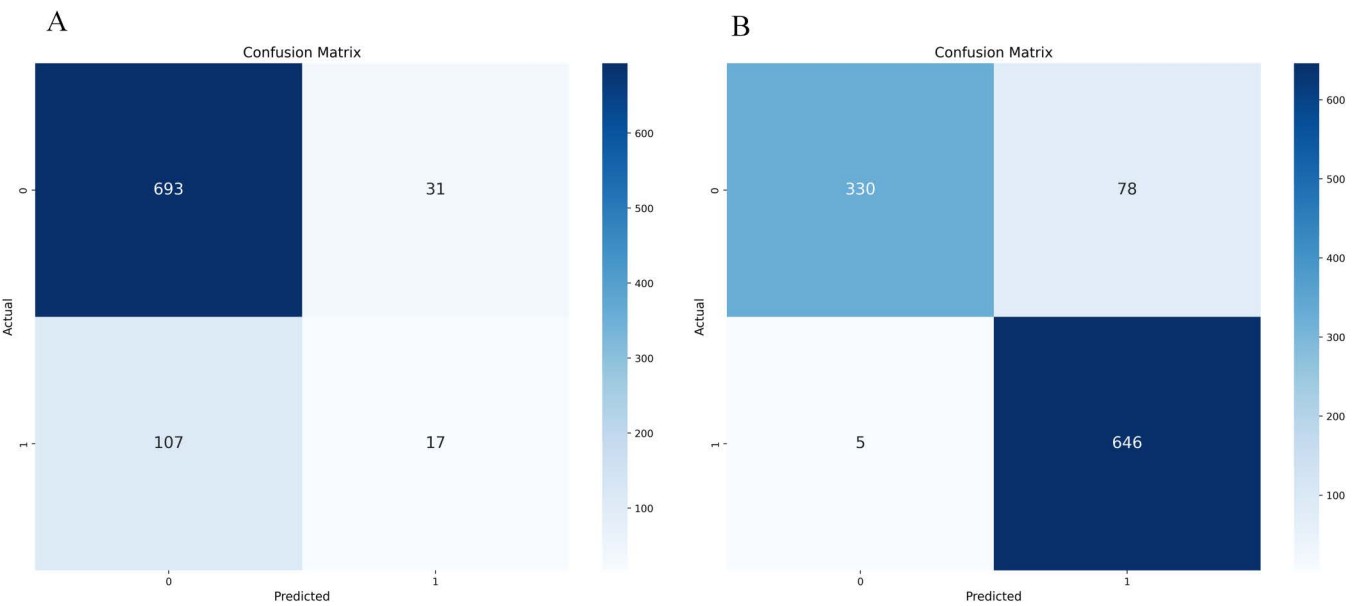

**Fig 6. Initialization confusion matrix of KNN (with and without SMOTE-ENN). (a)** Without. **(b)** With SMOTE-ENN.

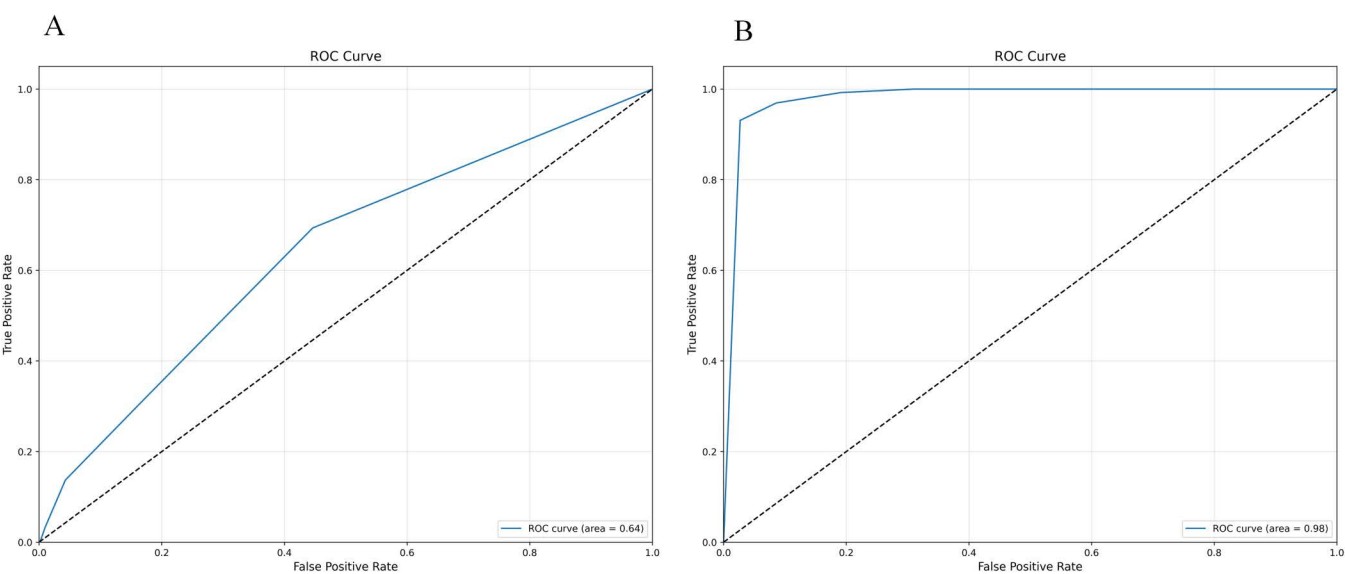

**Fig 7. Initialization ROC Plot of KNN (with and without SMOTE-ENN). (a)** Without. **(b)** With SMOTE-ENN.

indicating that precision is a critical metric for evaluating SVM performance on imbalanced datasets, as shown in Table 10. Among them, the kernel of SVM uses rbf, with a Degree of 3 and C of 1.0.

 Whether the initial model uses the confusion matrix and ROC curve displayed by SMOTE-ENN, as shown in Figs 8–9.

 With SMOTE-ENN, after optimising the model with grid search and processing the data with principal component analysis (PCA), the grid search optimised model outperforms PCA. After grid search optimisation, the model's predicted

**Table 9. Evaluation index after KNN optimization parameters.**

| Method | Accuracy | Precision | Recall | F1-score | AUC |
|---|---|---|---|---|---|
| Grid search | 95% | 92.84% | 99.54% | 96.07% | 0.98 |
| PCA | 94.26% | 91.68% | 98.30% | 94.88% | 0.98 |

**Table 10. Comparison of methods for balancing data in SVM.**

| Balance Method | Accuracy | Precision | Recall | F1_score | AUC |
|---|---|---|---|---|---|
| Without | 85.50% | 60% | 2.42% | 4.65% | 0.62 |
| SMOTE | 72.74% | 69.81% | 75.51% | 72.55% | 0.80 |
| SMOTE-ENN | 81.68% | 80.51% | 92.63% | 86.14% | 0.87 |

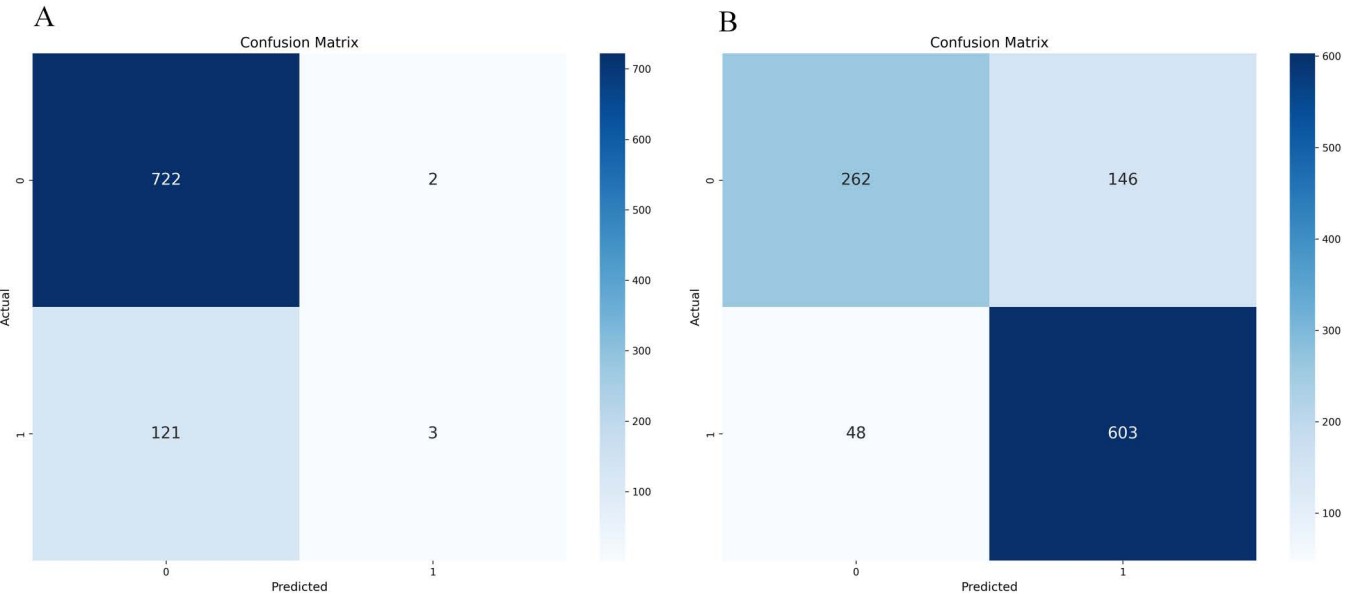

**Fig 8. Initialization confusion matrix of SVM (with and without SMOTE-ENN). (a)** Without, **(b)** With SMOTE-ENN.

accuracy is 87.82%, precision is 86.66%, recall is 94.78%, and F1-score is 90.54%, as shown in Table 11. The optimised results show that 87.82% of persons are accurately predicted, 94.78% of actual patients are correctly predicted to have the disease, and 5.22% of patients are mistakenly projected to not have the disease. Overall, as demonstrated in Table 4.8, the SVM model performs well on the coronary heart disease dataset.

In conclusion, after modifying the model, we saw considerable increases in four important metrics: correct identification of negative and positive classes increased, while wrong identification of negative as positive and positive as negative declined. This pattern implies that the overall model performance is improving. The attempt to reduce missed detections of coronary heart disease is particularly notable, as minimising the events of the positive class being misclassified as negative and the negative class being misclassified as positive also helps to avoid misdiagnosis. These performance enhancements are predicted to boost overall accuracy and F1 score, both of which are important metrics for assessing classifier performance.

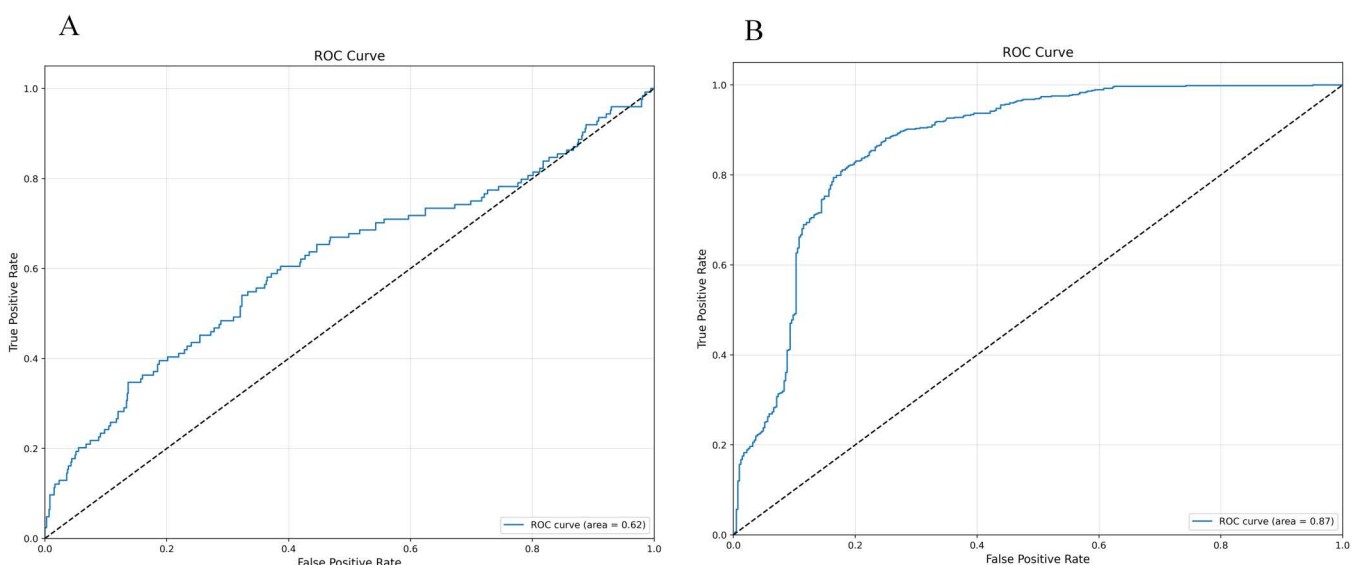

**Fig 9. Initialization ROC Plot of SVM (with and without SMOTE-ENN). (a)** Without, **(b)** With SMOTE-ENN.

**Table 11. Evaluation index after SVM optimization parameters.**

| Method | Accuracy | Precision | Recall | F1-score | AUC |
|---|---|---|---|---|---|
| Grid search | 87.82% | 86.66% | 94.78% | 90.54% | 0.92 |
| PCA | 81.66% | 81.40% | 88.01% | 84.57% | 0.88 |

**Table 12. Comparison of methods for balancing data in XGBoost.**

| Balance Method | Accuracy | Precision | Recall | F1_score | AUC |
|---|---|---|---|---|---|
| Without | 83.02% | 30.77% | 12.90% | 18.18% | 0.64 |
| SMOTE | 88.80% | 90.70% | 85.28% | 87.90% | 0.95 |
| SMOTE-ENN | 91.50% | 92.05% | 94.32% | 93.17% | 0.97 |

## E. Classification by gradient boosted trees

The use of SMOTE-ENN with XGBoost leads to a balanced performance across all metrics, with an AUC improvement from 0.64 to 0.97. This echoes the findings of Chen & Guestrin [47], who showed that boosting algorithms like XGBoost could benefit from balanced datasets, particularly in improving model AUC, as shown in Table 12. Among them, the n_estimators in XGBoost is 50, max_depth is 3, learning_rate is 0.01, and colsample_bytree is 0.98.

Whether the initial model uses the confusion matrix and ROC curve displayed by SMOTE-ENN, as shown in Figs 10–11

With SMOTE-ENN, after processing the data using both principle component analysis (PCA) and the grid search optimisation model, PCA's results outperform the grid search model's. Currently, as seen in Table 13. Among them, n_components in PCA is 8.

Following dimensionality reduction by PCA, the accuracy of the XGBoost model is 94.14%, indicating that 94.14% of the instances are accurately predicted by the XGBoost model. High accuracy is crucial for medical diagnosis, but other metrics must also be taken into account for a complete performance assessment. The accuracy stands for the percentage

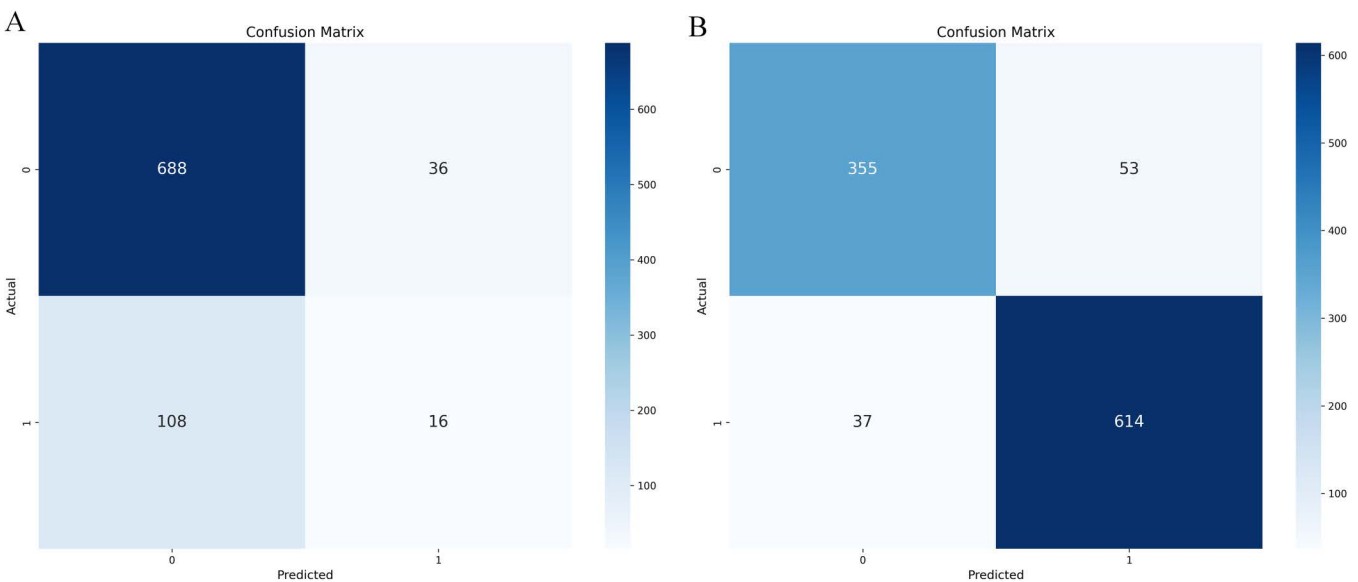

**Fig 10. Initialization confusion matrix of XGBoost (with and without SMOTE-ENN). (a)** Without, **(b)** With SMOTE-ENN.

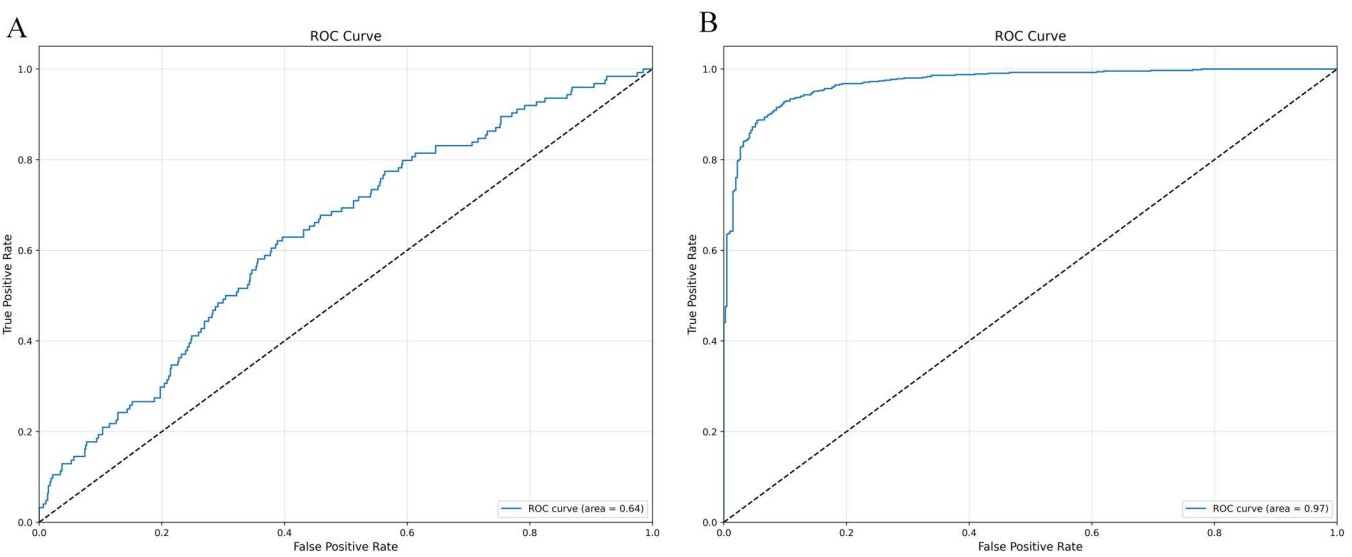

**Fig 11. Initialization ROC Plot of XGBoost (with and without SMOTE-ENN). (a)** Without, **(b)** With SMOTE-ENN.

of all cases, or 92% of them, in which the model correctly predicts a positive class (having coronary heart disease). With a 92% accuracy rate, there is a good chance that a patient will actually have coronary heart disease when the model makes this prediction. Recall percentage is 97.66%. The model can correctly forecast the percentage of positive class among all occurrences of genuine positive class (truly suffering from coronary heart disease). This suggests that the model is extremely effective at detecting actual cases of coronary heart disease. The model achieves an excellent compromise between not missing true instances and reducing misdiagnosis with an F1 score of 94.75%.

**Table 13. Evaluation index after XGBoost optimization parameters.**

| Method | Accuracy | Precision | Recall | F1-score | AUC |
|---|---|---|---|---|---|
| Grid search | 92.45% | 92.55% | 95.39% | 93.95% | 0.97 |
| PCA | 94.14% | 92% | 97.66% | 94.75% | 0.98 |

The model's performance indicators for predicting coronary heart disease are notably high, with an exceptional recall rate, crucial in medical diagnosis to minimize false negatives (FN). While false positives (FP) occur, their rate is relatively low, indicating good predictive reliability. High accuracy and F1 scores suggest strong overall model performance. However, the near-perfect AUC raises concerns about potential overfitting due to excessive optimization.

## F.Classification by Random Forest

Random Forest sees a rise in accuracy from 85.26% to 92.16% and in AUC from 0.68 to 0.98 with SMOTE-ENN. The study by Breiman [48] suggested that Random Forest inherently performs well with balanced datasets, and the application of SMOTE-ENN, as shown in the work of Blagus & Lusa [49], can effectively improve Random Forest performance by addressing class imbalance. as shown in Table 14. Among them, the n_estimators in RF is 100, min_samples_split is 2, and min_samples_leaf is 1.

Whether the initial model uses the confusion matrix and ROC curve displayed by SMOTE-ENN, as shown in Figs 12–13.

After the model has been improved using grid search, principal component analysis (PCA) is used to examine the data. The findings suggest that PCA outperforms the grid search-optimised model. At this point, n_components = 1. According to Table 15.

With SMOTE-ENN, following PCA, the model had an accuracy of 97.91% in determining whether individuals had coronary heart disease. This high accuracy usually reflects the model's good generalization ability, which means it can keep steady predicting capabilities on unseen data. With a precision of 98.13%, the model demonstrates excellent credibility in predicting individuals with genuine coronary heart disease, which is very relevant in the medical industry because it decreases the chance of misdiagnosing healthy people. The algorithm correctly identified virtually all real coronary heart disease patients, minimizing the chance of missed diagnoses, which is critical for disease prevention and therapy. Because the F1 score is an average of precision and recall, a score of 97.90% indicates a balanced performance in patient identification and avoidance of misdiagnosing healthy persons.

In summary, these evaluation indicators show that, following dimensionality reduction, the model performs brilliantly in dealing with coronary heart disease datasets, reliably identifying patients while minimising misdiagnosis and missing diagnosis. This is a useful feature for medical decision-making systems, particularly those used in the prevention and treatment of life-threatening disorders. However, in actual applications, clinical trials and professional judgements of doctors must be combined to evaluate the model's practicability completely.

The application of SMOTE-ENN consistently improves the performance of various machine learning algorithms when addressing imbalanced datasets. This enhancement is supported by extensive research highlighting SMOTE-ENN's effectiveness in resolving class imbalance issues across diverse machine learning contexts.

**Table 14. Comparison of methods for balancing data in Random Forest.**

| Balance Method | Accuracy | Precision | Recall | F1_score | AUC |
|---|---|---|---|---|---|
| Without | 85.26% | 47.37% | 7.26% | 12.58% | 0.68 |
| SMOTE | 89.64% | 88.52% | 89.94% | 89.23% | 0.97 |
| SMOTE-ENN | 92.16% | 90.57% | 97.39% | 93.85% | 0.98 |

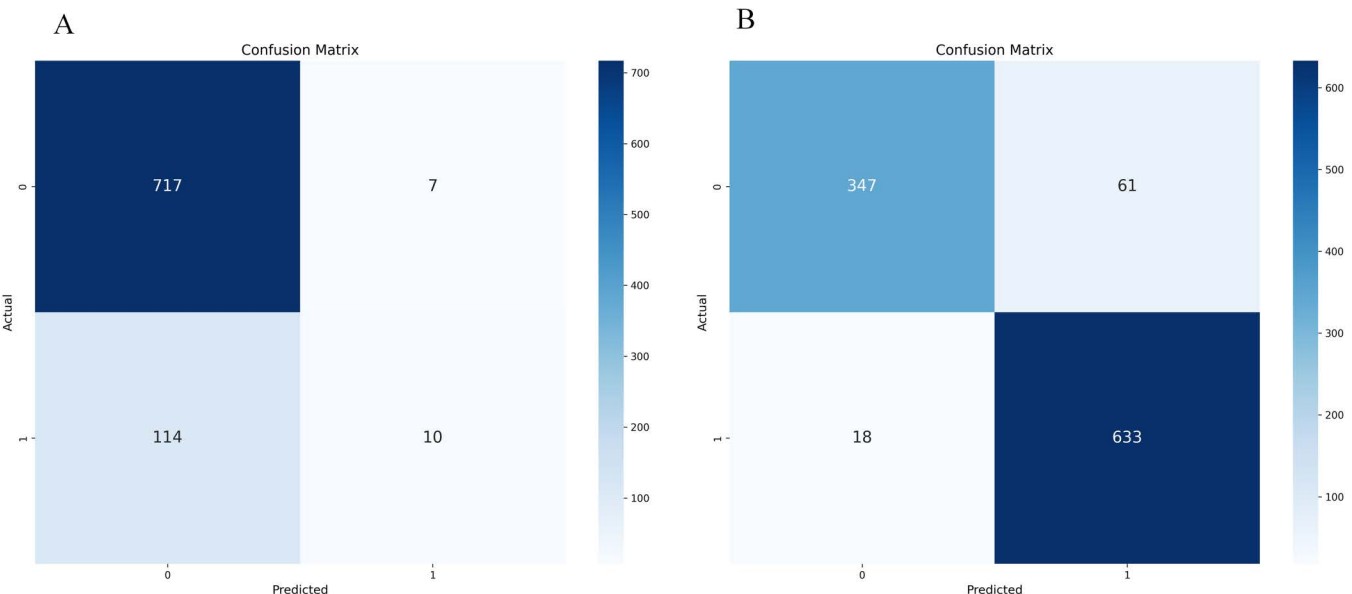

**Fig 12. Initialization confusion matrix of Random Forest (with and without SMOTE-ENN). (a) Without, (b) With SMOTE-ENN.**

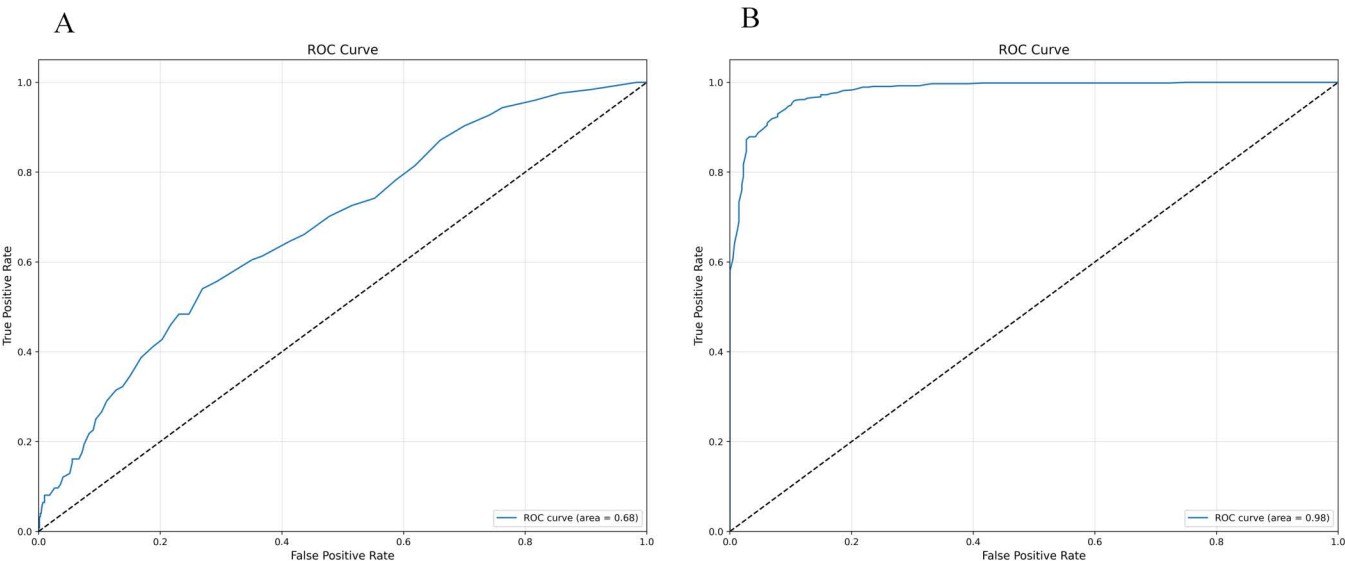

**Fig 13. Initialization confusion matrix of Random Forest (with and without SMOTE-ENN). (a)** Without, **(b)** With SMOTE-ENN.

**Table 15. Evaluation index after Random Forest optimization parameters.**

| Method | Accuracy | Precision | Recall | F1-score | AUC |
|---|---|---|---|---|---|
| Grid search | 91.88% | 90.07% | 97.54% | 93.66% | 0.98 |
| PCA | 97.91% | 96.95% | 98.83% | 97.88% | 0.98 |

To minimize prediction bias and improve practical performance, the dataset is first balanced using SMOTE-ENN. Subsequently, grid search and PCA are employed for parameter optimization and dimensionality reduction, further reducing prediction errors.

Among the evaluated models, Random Forest emerges as the best-performing algorithm after applying SMOTE-ENN and PCA. It achieves an AUC of 0.98, comparable to Decision Tree, KNN, and XGBoost, but with a higher F1-score. This high F1-score reflects the model's ability to effectively distinguish between patients and non-patients, minimizing false negatives and false positives.

In summary, Random Forest surpasses all other algorithms in terms of AUC, stability, and bias reduction. Its excellent performance across evaluation metrics, including a high 5-fold cross-validation accuracy score, underscores its stability and reliability, making it the most suitable choice for predicting coronary heart disease.

## H. Contrast Experiment

Through the comparative analysis of the performance of each model under the use of SMODS-ENN and the original imbalanced data (Without). According to Table 16, it can be clearly observed that the SMODS-ENN method effectively alleviates the overfitting phenomenon, while the models under the original imbalanced data generally have the risk of overfitting. Under the original data, although the model achieved a relatively high accuracy rate in the training set (such as 90.86% in RF training), its ability to recognize a few classes during testing was extremely poor (Recall was only 7.26%), indicating that the model overly relied on the majority class patterns in the training set and was unable to generalize to real scenarios. By synthesizing samples and cleaning noise, SMOTE-ENN enables the model to learn features more evenly during training. The Recall during testing generally exceeds 90%, and the performance during training and testing improves simultaneously, verifying its effectiveness in suppressing overfitting.

In conclusion, SMODS-ENN optimizes the adaptability of the model to the data distribution through data balance, enhancing the generalization ability while avoiding overfitting. However, the original data causes the model to fall into the local overfitting trap due to category bias.

Furthermore, the test results were segmented and tested according to gender, age and educational level, and the test results are shown in Table 17. First of all, there are certain differences in the performance of the KNN model among different genders. For males, the accuracy rate was 0.903, the precision rate was 0.884, the recall rate was 0.981, and the F1 score was 0.930. The relevant indicators of females performed slightly better, with an accuracy rate of 0.927, a precision rate of 0.889, a recall rate of 0.991, and an F1 score of 0.937. From this, it can be seen that the KNN model performs slightly better on female data and outperforms men in both recall rate and F1 score.

**Table 16. Evaluation results of different models using various data balancing method trees.**

| Model | Balance Method | Train-Accuracy | Test-Accuracy | Precision | Recall | F1_score | AUC |
|---|---|---|---|---|---|---|---|
| DT | Without | 82.35 | 74.76% | 20.78% | 25.81% | 23.02% | 0.54 |
| | SMOTE-ENN | 90.51 | 85.65% | 86.75% | 90.48% | 88.57% | 0.83 |
| KNN | Without | 02.56 | 83.73% | 35.42% | 13.71% | 19.77% | 0.64 |
| | SMOTE-ENN | 96.86 | 92.16% | 89.23% | 99.23% | 93.96% | 0.98 |
| SVM | Without | 93.75 | 85.50% | 60.00% | 2.42% | 4.65% | 0.62 |
| | SMOTE-ENN | 87.21 | 81.68% | 80.51% | 92.63% | 86.14% | 0.87 |
| XGBoost | Without | 91.76 | 83.02% | 30.77% | 12.90% | 18.18% | 0.64 |
| | SMOTE-ENN | 95.14 | 91.50% | 92.05% | 94.32% | 93.17% | 0.97 |
| RF | Without | 90.86 | 85.26% | 47.37% | 7.26% | 12.58% | 0.68 |
| | SMOTE-ENN | 97.34 | 92.16% | 90.57% | 97.39% | 93.85% | 0.98 |

From the perspective of age, the performance of the 30–50 age group was the most outstanding, with an accuracy rate of 0.918, a precision rate of 0.877, a recall rate of 1.000, and an F1 score of 0.935. This indicates that the prediction results of this age group are relatively accurate and have a high recall rate. The performance of the 50–70 age group was slightly inferior. Although its precision rate was 0.904, the recall rate was 0.977, and the F1 score also remained at a relatively high level (0.939). This might indicate that the model has slight deficiencies in recognizing certain features of the older group, but the overall performance is still acceptable. In terms of educational attainment, the KNN model performs relatively closely at different educational levels. However, for the group with a higher educational attainment (grades 3–4), the F1 score is 0.940, and for the group with a lower educational attainment (grades 1–2), the F1 score is 0.930. This indicates that the group with a higher educational level may be more in line with the recognition pattern of the model. This leads to its even more outstanding performance.

Overall, the KNN model shows certain differences in the segmentation of gender, age and educational level, especially prominent among women and younger groups. In the age group of 30–50 years old, the performance of the RF model was particularly outstanding, with an accuracy rate of 0.950, a precision rate of 0.939, and an F1 score of 0.959, demonstrating the advantages of the RF model in identifying data of this age group. In contrast, the RF model is stable and performs better than the KNN model in most cases, especially. Therefore, the RF model performs more evenly in most subdivision dimensions, especially among groups with higher age and education levels, showing higher accuracy and precision, and is applicable to a wide range of population distributions.

## Discussion

In this study, a CHD prediction model was established using authoritative public datasets, and it was proposed that SMOTE-ENN combined with PCA can further improve the performance of the CHD prediction model.

The TenyearCHD prediction model developed in this study holds significant potential for application in clinical practice. It can serve as a valuable tool for healthcare professionals in assessing the likelihood of patients developing cardiovascular disease within a ten-year period. By providing early risk stratification, the model enables physicians to identify high-risk individuals promptly, facilitating timely interventions and personalized treatment plans. This can lead to improved patient outcomes, more efficient allocation of healthcare resources, and cost-effective management of cardiovascular disease. Furthermore, the model's ability to predict TenyearCHD can aid in patient education and counseling, empowering individuals to make informed decisions about lifestyle modifications and adherence to preventive measures. Overall, the

**Table 17. Prediction results of KNN and RF under different attribute characteristics.**

| Model | Attribute | Details | Accuracy | Precision | Recall | F1_score |
|---|---|---|---|---|---|---|
| KNN | Male | Man | 0.903 | 0.884 | 0.981 | 0.930 |
| | | Woman | 0.927 | 0.889 | 0.991 | 0.937 |
| | Age | 30~50 | 0.918 | 0.877 | 1.000 | 0.935 |
| | | 50~70 | 0.917 | 0.904 | 0.977 | 0.939 |
| | Education | 1-2 | 0.909 | 0.874 | 0.993 | 0.930 |
| | | 3-4 | 0.923 | 0.887 | 1.000 | 0.940 |
| RF | Male | Man | 0.913 | 0.905 | 0.970 | 0.936 |
| | | Woman | 0.906 | 0.890 | 0.951 | 0.922 |
| | Age | 30~50 | 0.950 | 0.939 | 0.979 | 0.959 |
| | | 50~70 | 0.924 | 0.917 | 0.973 | 0.944 |
| | Education | 1-2 | 0.918 | 0.900 | 0.974 | 0.936 |
| | | 3-4 | 0.923 | 0.886 | 1.000 | 0.940 |

integration of this machine learning prediction model into clinical workflows can enhance the prevention and management of cardiovascular disease, contributing to better public health.

This study employed the Kaggle public dataset, which has been widely utilized by numerous researchers. However, the dataset has several limitations. First, the dataset contains a high prevalence of missing values, which can significantly affect model performance. Second, there is an imbalance in the distribution of various characteristics, such as smaller samples of certain educational backgrounds or age groups. These issues collectively result in insufficient accuracy of the TenyearCHD prediction model. Additionally, the dataset lacks data on high-density lipoprotein cholesterol (HDL-C), which prevents comparison with the Framingham Risk Score and use of the ASCVD risk calculator. Therefore, future work will focus on using a more comprehensive dataset that includes diverse racial and regional populations. Furthermore, HDL-C data will be collected to develop a high-precision HDL-C prediction model, enabling the full utilization of other parameter information for calculating the Framingham Risk Score.

Finally, deep learning technology has demonstrated excellent performance when dealing with complex and large-scale data. Therefore, in CHD risk prediction, through automated feature extraction and nonlinear relationship modeling, it can improve the accuracy and sensitivity of prediction to a certain extent. In addition, data collection based on wearable devices makes real-time monitoring possible. By integrating sensor technology and deep learning models, it is possible to assess an individual's health status more accurately and predict potential disease risks in a timely manner. Consider embedding these advanced models into micro-medical devices, which will provide portable CHD detection tools for relevant personnel. By integrating deep learning models with wearable devices, users can monitor their cardiovascular health at any time in their daily lives, achieving the goals of early warning and personalized intervention, thereby effectively reducing the risk of heart attacks and improving public health.

## Conclusion

This study explored the effectiveness of various machine learning models for predicting coronary heart disease (CHD) while addressing data imbalance and bias through SMOTE-ENN and PCA techniques. Five models—Decision Tree, KNN, SVM, XGBoost, and Random Forest—were evaluated using key performance metrics such as accuracy, precision, recall, F1-score, and AUC.The application of SMOTE-ENN successfully mitigated class imbalance in the dataset, improving model performance. PCA further enhanced prediction accuracy by reducing dimensionality and optimizing model parameters. Among the models tested, the Random Forest classifier demonstrated the best performance, achieving 92.16% accuracy, 90.57% precision, 93.39% recall, 93.85% F1-score, and an AUC of 0.98.

## Author contributions

**Conceptualization:** Xinyi Wei, Boyu Shi.

**Data curation:** Xinyi Wei, Boyu Shi.

**Methodology:** Xinyi Wei.

**Writing – original draft:** Xinyi Wei.

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
