## [Decision Letter · Decision Letter 0]

15 Apr 2025

PONE-D-25-04242Reducing Bias In Coronary Heart Disease prediction using smote-enn and pcaPLOS ONE

Dear Dr. Wei,

Thank you for submitting your manuscript to PLOS ONE. After careful consideration, we feel that it has merit but does not fully meet PLOS ONE’s publication criteria as it currently stands. Therefore, we invite you to submit a revised version of the manuscript that addresses the points raised during the review process.

We look forward to receiving your revised manuscript.

Kind regards,

Li-Da Wu

Academic Editor

PLOS ONE

**Journal Requirements:**

1. When submitting your revision, we need you to address these additional requirements. Please ensure that your manuscript meets PLOS ONE's style requirements, including those for file naming. The PLOS ONE style templates can be found at https://journals.plos.org/plosone/s/file?id=wjVg/PLOSOne_formatting_sample_main_body.pdf and https://journals.plos.org/plosone/s/file?id=ba62/PLOSOne_formatting_sample_title_authors_affiliations.pdf 2. Please note that PLOS ONE has specific guidelines on code sharing for submissions in which author-generated code underpins the findings in the manuscript. In these cases, we expect all author-generated code to be made available without restrictions upon publication of the work. Please review our guidelines at https://journals.plos.org/plosone/s/materials-and-software-sharing#loc-sharing-code and ensure that your code is shared in a way that follows best practice and facilitates reproducibility and reuse. 3. Please provide a complete Data Availability Statement in the submission form, ensuring you include all necessary access information or a reason for why you are unable to make your data freely accessible. If your research concerns only data provided within your submission, please write "All data are in the manuscript and/or supporting information files" as your Data Availability Statement. 4. PLOS requires an ORCID iD for the corresponding author in Editorial Manager on papers submitted after December 6th, 2016. Please ensure that you have an ORCID iD and that it is validated in Editorial Manager. To do this, go to ‘Update my Information’ (in the upper left-hand corner of the main menu), and click on the Fetch/Validate link next to the ORCID field. This will take you to the ORCID site and allow you to create a new iD or authenticate a pre-existing iD in Editorial Manager.

**Additional Editor Comments:**

I recommend major revision of this manuscript based on comments of the reviewers.

Reviewers' comments:

Reviewer's Responses to Questions

**Comments to the Author**

1. Is the manuscript technically sound, and do the data support the conclusions?

Reviewer #1: Yes

Reviewer #2: Partly

Reviewer #3: Partly

2. Has the statistical analysis been performed appropriately and rigorously? 

Reviewer #1: Yes

Reviewer #2: No

Reviewer #3: Yes

3. Have the authors made all data underlying the findings in their manuscript fully available?

Reviewer #1: Yes

Reviewer #2: Yes

Reviewer #3: No

4. Is the manuscript presented in an intelligible fashion and written in standard English?

Reviewer #1: Yes

Reviewer #2: Yes

Reviewer #3: Yes

5. Review Comments to the Author

**Reviewer #1:**  This study proposes a machine learning framework for coronary heart disease (CHD) prediction using SMOTE-ENN for data balancing and PCA for feature optimization. The research integrates various machine learning models, including Random Forest, SVM, KNN, XGBoost, and Decision Trees, to improve prediction accuracy. While the study presents an interesting approach, several major revisions are required to enhance the clarity, methodological transparency, and practical applicability of the findings.

Key Concerns & Required Revisions

Lack of Clinical Validation and Benchmarking Against Standard CHD Risk Scores

The study lacks a comparison with established CHD risk models such as the Framingham Risk Score or ASCVD risk calculator.

It is unclear how the proposed machine learning models outperform traditional risk prediction models in clinical utility.

The authors should include a direct comparison of model performance against these standard clinical scores.

Data Imbalance and Model Interpretability

While SMOTE-ENN is applied to handle data imbalance, it is not clear whether over-sampling led to overfitting, particularly in models like Random Forest and XGBoost.

The study should include a sensitivity analysis on the impact of different resampling techniques (e.g., SMOTE, SMOTE-Tomek) on model performance.

Generalizability and External Validation

The study is based on a dataset from Framingham/Kaggle, which does not represent a globally diverse population.

Did the authors attempt external validation in an independent cohort? If not, this should be addressed as a major limitation.

The authors should discuss potential biases in CHD risk prediction across different ethnicities, ages, and socioeconomic backgrounds.

Statistical & Methodological Clarity

How were hyperparameters optimized? The methodology mentions grid search, but specific parameters for each model should be detailed.

How was missing data handled? Were imputation techniques used, or were missing values removed?

What statistical tests were used to compare model performances? The authors should clarify if paired t-tests or DeLong’s test for AUC comparison were applied.

Clinical Relevance & Implementation

The manuscript does not discuss how the model would be integrated into clinical practice.

The authors should provide decision thresholds for CHD risk classification (e.g., at what probability cutoff should an individual be flagged as high risk?).

Consider discussing how these models can be incorporated into real-world CHD screening workflows.

Discussion on Alternative Approaches to CHD Risk Prediction

The authors focus solely on data-driven methods but should discuss other potential approaches such as genetic risk scores, deep learning models, and wearable device-based risk prediction.

**Reviewer #2: ** Review:

Some explanations, such as dataset description and missing data handling, could be more concise and precise.

Clarifying the rationale behind certain choices, like encoding methods and feature selection, would enhance readability.

The discussion on standardization, normalization, and PCA could be expanded slightly for better clarity.

Minor inconsistencies in terminology and figure references should be addressed for a smoother flow.

Explaining what would be the effect and impact i.e. how it can be employed in clinical practice will add into the value of your manuscript

**Reviewer #3:**  I acknowledge the effort and dedication put into this work and appreciate the opportunity to review it. Kindly take note of the following;

1. Kindly indicate all your in-text references or citations as superscripts or preferably in square bracket [...] as recommended by PLOS ONE to differentiate it from the main text.

2. The in-text reference in Line 45, 46 and 53 are quite misleading. It is best if you separate them with commas (,).

3. Check the word "practiceto" in Line 51.

4. Line 53 - There is an error information "Error! Reference source not found." Kindly fix that.

5. The citation "(Kaggle, n.d.)" introduces another reference which violates PLOS ONE's citation recommendation. Moreover, there is no date.

6. Line 146 - the statement is not clear

7. Table 6 is difficult to follow. Kindly format it well

8. All the figures in the work are blurry and difficult to read from. Kindly provide a sharper image which can easily be referred to

9. The work needs major reorganization in order to clearly identify the methods used to assess the various machine learning models for predicting coronary heart disease, the results and the discussion with adequate synthesis of existing literature

6. PLOS authors have the option to publish the peer review history of their article (what does this mean? ). If published, this will include your full peer review and any attached files.

**Do you want your identity to be public for this peer review?** For information about this choice, including consent withdrawal, please see our Privacy Policy .

Reviewer #1: No

Reviewer #2: No

Reviewer #3: **Yes: ** Isaac Boateng

---

## [Author Response · Author response to Decision Letter 1]

12 May 2025

Dear Reviewer and Editor,

Regarding this section, I have uploaded the reviewers' comments to the attached files. Please refer to the document titled "Response to reviewers.DOCX."

Thank you.

---

## [Decision Letter · Decision Letter 1]

18 Jun 2025

Reducing Bias In Coronary Heart Disease Prediction Using Smote-enn And Pca

PONE-D-25-04242R1

Dear Dr. Wei,

We’re pleased to inform you that your manuscript has been judged scientifically suitable for publication and will be formally accepted for publication once it meets all outstanding technical requirements.

Kind regards,

Li-Da Wu

Academic Editor

PLOS ONE

Additional Editor Comments (optional):

Reviewers' comments:

Reviewer's Responses to Questions

**Comments to the Author**

1. If the authors have adequately addressed your comments raised in a previous round of review and you feel that this manuscript is now acceptable for publication, you may indicate that here to bypass the “Comments to the Author” section, enter your conflict of interest statement in the “Confidential to Editor” section, and submit your "Accept" recommendation.

Reviewer #3: All comments have been addressed

Reviewer #4: All comments have been addressed

Reviewer #5: (No Response)

2. Is the manuscript technically sound, and do the data support the conclusions?

Reviewer #3: Yes

Reviewer #4: Yes

Reviewer #5: No

3. Has the statistical analysis been performed appropriately and rigorously? 

Reviewer #3: Yes

Reviewer #4: Yes

Reviewer #5: No

4. Have the authors made all data underlying the findings in their manuscript fully available?

Reviewer #3: No

Reviewer #4: Yes

Reviewer #5: No

5. Is the manuscript presented in an intelligible fashion and written in standard English?

Reviewer #3: Yes

Reviewer #4: Yes

Reviewer #5: No

6. Review Comments to the Author

Reviewer #3: The authors have demonstrated diligent attention to the reviewer feedback, implementing all suggested improvements and providing clear justifications where modifications were not made. The revised manuscript successfully addresses the concerns raised in the initial review

Reviewer #4: The authors have thoroughly considered the reviewers’ comments and incorporated the suggestions where relevant and appropriate. The revised manuscript reflects substantial improvements and is now well-positioned for publication. It is in a suitable form to be made available to readers.

Reviewer #5: The manuscript does not meet the criteria for publication in a high-quality journal primarily due to the absence of comparisons with established CHD risk models, which is a critical oversight for a study claiming to advance CHD prediction. This lack of benchmarking, combined with limited generalizability, absence of external validation, potential overfitting concerns, superficial discussion of clinical integration, and incomplete exploration of alternative approaches, significantly weakens the study’s scientific contribution. Additionally, methodological and presentation issues noted by reviewers further undermine the manuscript’s quality. While the authors have made efforts to address reviewer comments, the fundamental limitations, particularly the lack of comparison with standard clinical models, cannot be overlooked. Therefore, I recommend rejecting the manuscript for publication.

7. PLOS authors have the option to publish the peer review history of their article (what does this mean? ). If published, this will include your full peer review and any attached files.

**Do you want your identity to be public for this peer review?** For information about this choice, including consent withdrawal, please see our Privacy Policy .

Reviewer #3: **Yes: ** Isaac Boateng

Reviewer #4: **Yes: ** Muhammad Abdus Salam

Reviewer #5: No

---

## [Editor Report · Acceptance letter]

PONE-D-25-04242R1

PLOS ONE

Dear Dr. Wei,

I'm pleased to inform you that your manuscript has been deemed suitable for publication in PLOS ONE. Congratulations! Your manuscript is now being handed over to our production team.

Kind regards,

on behalf of

Dr. Li-Da Wu

Academic Editor

PLOS ONE